# The Effect of Non-Invasive Brain Stimulation (NIBS) on Executive Functioning, Attention and Memory in Rehabilitation Patients with Traumatic Brain Injury: A Systematic Review

**DOI:** 10.3390/diagnostics11040627

**Published:** 2021-03-31

**Authors:** Takatoshi Hara, Aturan Shanmugalingam, Amanda McIntyre, Amer M. Burhan

**Affiliations:** 1Department of Rehabilitation Medicine, Jikei University School of Medicine, Tokyo 105-8461, Japan; t_hara1019@jikei.ac.jp; 2St. Joseph’s Health Care, Parkwood Institute Mental Health, London, ON N6A 4V2, Canada; Aturan.Shanmugalingam@sjhc.london.on.ca; 3Parkwood Institute Research, Parkwood Institute, London, ON N6C 0A7, Canada; Amanda.McIntyre@sjhc.london.on.ca; 4Department of Psychiatry and Medicine, Schulich School of Medicine and Dentistry, Western University, London, ON N6A 5C1, Canada; 5Ontario Shores Centre for Mental Health Sciences, Whitby, ON L1N 5S9, Canada; 6Department of Psychiatry, University of Toronto, Toronto, ON M5T 1R8, Canada

**Keywords:** traumatic brain injury, non-invasive brain stimulation, transcranial magnetic stimulation, transcranial direct current stimulation

## Abstract

In recent years, the potential of non-invasive brain stimulation (NIBS) for therapeutic effects on cognitive functions has been explored for populations with traumatic brain injury (TBI). However, there is no systematic NIBS review of TBI cognitive impairment with a focus on stimulation sites and stimulation parameters. The purpose of this study was to conduct a systematic review examining the effectiveness and safety of NIBS for cognitive impairment after a TBI. This study was prospectively registered with the PROSPERO database of systematic reviews (CRD42020183298). All English articles from the following databases were searched from inception up to 31 December 2020: Pubmed/MEDLINE, Scopus, CINAHL, Embase, PsycINFO and CENTRAL. Randomized and prospective controlled trials, including cross-over studies, were included for analysis. Studies with at least five individuals with TBI, whereby at least five sessions of NIBS were provided and used standardized neuropsychological measurement of cognition, were included. A total of five studies met eligibility criteria. Two studies used repetitive transcranial magnetic stimulation (rTMS) and three studies used transcranial direct current stimulation (tDCS). The pooled sample size was 44 individuals for rTMS and 91 for tDCS. Three of five studies combined cognitive training or additional therapy (computer assisted) with NIBS. Regarding rTMS, target symptoms included attention (*n* = 2), memory (*n* = 1), and executive function (*n* = 2); only one study showing significant improvement compared than control group with respect to attention. In tDCS studies, target symptoms included cognition (*n* = 2), attention (*n* = 3), memory (*n* = 3), working memory (WM) (*n* = 3), and executive function (*n* = 1); two of three studies showed significant improvement compared to the control group with respect to attention and memory. The evidence for NIBS effectiveness in rehabilitation of cognitive function in TBI is still in its infancy, more studies are needed. In all studies, dorsolateral prefrontal cortex (DLPFC) was selected as the stimulation site, along with the stimulation pattern promoting the activation of the left DLPFC. In some studies, there was a significant improvement compared to the control group, but neither rTMS nor tDCS had sufficient evidence of effectiveness. To the establishment of evidence we need the evaluation of brain activity at the stimulation site and related areas using neuroimaging on how NIBS acts on the neural network.

## 1. Introduction

Traumatic brain injury (TBI) is a major cause of death and chronic disability worldwide, particularly for young and elderly patients. Each year an estimated 69 million individuals will suffer a TBI, the vast majority of which will be mild (81%) and moderate (11%) in severity [1].

One of the many consequences of TBI is often cognitive impairment, which may lead to significant dysfunction. In a national epidemiological cohort study of population and prevalence after brain injury in the chronic phase, Nakajima et al. [2] reported that the most common cognitive symptoms were memory impairment (90%), attention disorder (82%), and executive function impairment (75%). TBI is largely heterogeneous, with cases often presenting differently despite seemingly similar injuries. The scope and severity of cognitive symptoms depend on many factors, including injury mechanism and severity, demographic, and social factors. The TBI Model Systems National Database reported that 23.7% of moderate and severe TBI patients (older than 16 years) who received inpatient rehabilitation demonstrated improvement in their cognition within 5 years; 24% of the sample reported cognitive decline [3]. This illustrates that cognitive impairment may persist beyond the acute phase of recovery. Further, these cognitive issues cause significant functional limitations including rehabilitation effort, resuming work, and the need for additional support [4,5].

Cognitive rehabilitation is the mainstay of treatment for cognitive deficits associated with TBI [6]. Cognitive rehabilitation typically focuses on compensatory strategies to improve an individual’s functioning. Cognitive rehabilitation also facilitates learning. There is some literature demonstrating cognitive rehabilitation having the potential to improve cognitive function directly; however, the evidence for this effect is not sufficient [7]. Several systematic reviews have been reported on rehabilitation interventions for cognition in TBI [8,9]. However, as mentioned above, the effects are reported to be limited or controversial. For example, Chung et al. identified an insufficient amount of high-quality evidence to reach any generalized conclusions about the effect of cognitive rehabilitation on executive function [9].

Recently, the role of non-invasive brain stimulation (NIBS) in the rehabilitation of cognitive impairment in TBI has attracted significant attention [10]. In general, NIBS techniques use electrical and/or magnetic energy to induce change in excitability of the underlying brain cortex in a non-invasive fashion and potentially induce long-lasting neuroplastic changes. Although there are several methods for NIBS, rTMS and tDCS are currently the mainstream stimulation methods in clinical application [11,12]. Both rTMS and tDCS have been applied in the field of psychiatric disorders and especially to treat depression [13,14]. In recent years, the potential of NIBS to have therapeutic effects on cognitive function has been explored for brain injury populations as well [15,16,17]. We have previously reported a case in which improvement of cognitive deficits following brain injury was achieved by using rTMS combined with intensive rehabilitation [16]. Furthermore, the use of single photon emission computer tomography demonstrated changes in perfusion in the rTMS target sites and areas surrounding the targets [16]. Given the heterogeneous nature of cognitive impairment after a TBI, it is a difficult task to determine specific stimulation sites, stimulation parameters, and stimulation duration. In fact, in our case report, different stimulation sites and parameters were individually selected from the images and symptoms before stimulation [16]. NIBS could be a promising complementary treatment when used in combination with conventional cognitive rehabilitation to enhance rehabilitation in patients with brain injury; however, there is currently, no systematic review examining its effects on cognitive impairment after TBI. There are reviews of rTMS and tDCS in TBI [18,19,20]. Some reviews focused on depression, dizziness, central pain, and visual neglect [18,19], while others were not specifically focused on current paradigms of NIBS therapeutic application [19,20]. Therefore, in this study we conducted a systematic review focusing on the effectiveness and safety of therapeutic NIBS paradigms for cognitive impairment after a TBI with respect to memory, attention, and executive function. Additionally, we evaluated stimulation sites, stimulation parameters, neuropsychological tests, and secondary evaluations.

## 2. Materials and Methods

A protocol to review NIBS among individuals with stroke and TBI was prospectively registered with the PROSPERO database of systematic reviews (CRD42020183298); for clarity, we have opted to present the results for individuals with stroke and TBI separately.

### 2.1. Literature Search Strategy

The following sources were searched from inception and up to 31 December 2020 for literature published in the English language: Pubmed/MEDLINE, Scopus, CINAHL, Embase, PsycINFO and CENTRAL. Selected keywords included Acquired brain injury, Traumatic brain injury, Brain injury, Head injury, Craniocerebral trauma, Non-invasive brain stimulation, Transcranial magnetic stimulation, Theta-burst stimulation, Quadripluse stimulation, Transcranial Electrical Stimulation, Transcranial direct-current stimulation, Transcranial Alternating current stimulation, Cognition, Memory, Attention, Executive functioning. Post-concussion symptoms after TBI was excluded in this review. Variations of keywords were individualized for each scientific database. All retrieved articles were reviewed to ensure relevant articles were included for data synthesis. An example search strategy has been illustrated for Pubmed/MEDLINE in Figure 1.

### 2.2. Study Selection

Articles reporting on randomized and prospective controlled trials (RCT and PCT, respectively) were included for review. We included studies in which NIBS was used for cognitive rehabilitation or training for TBI, reported cognitive function pre- and post-intervention and included a minimum of five daily sessions of NIBS. Articles reporting on protocols, in-progress trials, retrospective studies or case reports were excluded. We included studies reporting on at least five TBI patients, who were 18–85 years old, and that provided at least 5 sessions over 5 days given that long-term cognitive improvement is likely related to the number of stimulation session/days, with more stimulation sessions resulting in a longer-lasting response [17].

Two authors (T.H. and A.S.) independently reviewed all potential studies for inclusion against the eligibility criteria. They examined the title and abstract and, where necessary, the full text of studies to assess if they were eligible for inclusion. If they could not reach agreement by discussion, a third author (A.B.) made the final decision about eligibility.

### 2.3. Data Extraction and and Synthesis

Two authors (T.H. and A.S.) independently used a standard form to extract study characteristics and outcome data from the studies. Discrepancies were checked against the original data. A third author (A.B.) made the final decision in the cases of disagreement. Data extracted from each study included author, year, sample size, sex, age, time between onset and treatment, target symptom, stimulation site, each NIBS parameter, rehabilitation, outcome measures, results, and safety reports. Several studies evaluated symptoms of cognitive impairment after TBI (i.e., attention, memory, working memory, executive functioning and cognition) using neuropsychological tests. Data were categorized by cognitive impairment symptom at the data extraction stage. However, some neuropsychological tests assessed more than one symptom simultaneously; in those cases, classification by symptoms of cognitive impairment was based on the author’s primary objective.

### 2.4. Methodological Quality

In studies assessing NIBS for TBI, some trials have examined a wide range of symptoms, such as upper and lower limbs, aphasia and spasticity. Additionally, some techniques combining multiple rehabilitation interventions have also been tried [11,15,16,21,22]. With this background, NIBS for cognitive impairment is also considered as one of the complimentary rehabilitation methods. Therefore, the Physiotherapy Evidence Database (PEDro) scoring system [23] was chosen to assess methodological quality of our selected studies. PEDro is widely used in systematic reviews in the rehabilitation area, and the PEDro tool has been used to score over 46,000 RCTs across 14 physiotherapy areas including a significant number in neurorehabilitation. PEDro assesses 11 areas of study quality that are answered with a “yes” (score = 1) or “no” (score = 0). The first item is a measure of external validity and is not used in calculating the final score (i.e., sum of items 2–11). Based on this assessment, all studies were given a Level of Evidence (LoE) according to a modified Sackett Scale [24].

## 3. Results

### 3.1. Study Selection

We identified 251 records through the searches after removal of duplicates. No additional records from other sources were identified. After screening the titles and abstracts, we excluded 244 records mainly because the studies were animal studies, abstracts only, articles reporting on protocols, in-progress trials, retrospective studies or case reports, systematic review, non-English language publications and completely irrelevant articles. Among those remaining (*n* = 6), three studies used repetitive transcranial magnetic stimulation (rTMS) and three studies used transcranial direct current stimulation (tDCS) [25,26,27,28,29,30]; however, one study [25] had less than 5 sessions or 5 days of NIBS and was excluded, leaving five studies for review [26,27,28,29,30].

### 3.2. Study Characteristics

The details of each study are provided in Table 1. For rTMS, the pooled sample size was 44 individuals with a range of 7 to 17 subjects per group. All articles studying rTMS were RCTs. The age range of the intervention group was 29.0–42.4 years, and for the control group, 32.6–41.3 years. The time between onset and treatment ranged from 3.9 to 18.3 months post TBI. For tDCS, the pooled sample size was 91 individuals with range of 11 to 16 subjects per group. All articles studying tDCS were RCTs. The age range of the intervention group was 29.2–37.7 years, and for the control group, 28.2–35.5 years. The time between onset of symptoms and treatment ranged from 41.1 days to 18.0 months post TBI.

Four studies were ranked as Level 1 evidence and one study as Level 2 evidence. All subjects were randomly allocated to groups appropriately. In all studies, intervention and control groups were similar at baseline regarding the most important prognostic indicators. Blinding was highly variable among studies. All studies yielded at least one important outcome measure from more than 85% of the subjects initially assigned to a group. In addition, except for one study, the results of statistical comparisons between groups and the presentation of point measures and measures of variability were adequately performed.

Three of five studies combined cognitive training or additional therapy (e.g., computer-assisted training) with NIBS [27,28,30]. Regarding assessment of cognitive impairment, all articles reported on attention [26,27,28,29,30], four articles reported on memory [26,28,29,30], three articles reported on WM [28,29,30] and executive function [26,27,29], and two articles reported on cognition measured by global measures like the minimental state exam (MMSE). In the rTMS study, there was one study each for excitatory and inhibitory stimulation [30]. All three articles involving tDCS used anodal simulation. Three articles performed a follow-up assessment after stimulation [26,28,30].

### 3.3. Outcomes

Table 2 shows study treatment characteristics, outcome measures and results for each study, for both rTMS and tDCS.

Effect of rTMS: Two studies evaluated the use of rTMS for cognitive impairment [26,27]. Neville et al. [26] used 10 Hz (excitatory) stimulation to the left dorsolateral prefrontal cortex (DLPFC) [26]. They measured attention, memory, and executive function before and after intervention, and at follow up at 90 days between the treatment and control group. The treatment group noted a significant improvement after 90 days in executive function but this was not significant compared to the control group. No significant differences were observed on the other neuropsychological tests. In the second study, Lee et al. [27] used 1 Hz (inhibitory) stimulation to the right DLPFC. They evaluated attention using the Trail making test (TMT) and executive function using the Stroop Color Word Test (SCWT). They reported significant improvement in attention compared to the control group.

#### 3.3.1. Effect of tDCS

Regarding tDCS, two studies assessed overall cognition as the outcome [29,30] all three trials [28,29,30] assessed attention, memory and working memory, and one trial [29] assessed executive function. One trial [28] used bilateral DLPFC as the stimulation site and two trials [29,30] used the left DLPFC. All trials involving tDCS used an anodal pattern. Sacco et al. [28] used anodal tDCS on the bilateral DLPFC and reported that the intervention group significantly improved in divided attention on the attention task of Repeatable Battery for the Assessment of the Neuropsychological Status (RBANS) [28]. No significant improvement was observed on the memory component of RBANS. In addition, Sacco et al. [28] examined functional Magnetic Resonance Imaging (fMRI) before and after the intervention during divided attention tasks and showed that brain activity was decreased in the right superior temporal gyrus (BA 42), right and left middle frontal gyrus (BA 6), right postcentral gyrus (BA 3) and left inferior frontal gyrus (BA 9). They indicated that such neural changes were normalization of previously abnormal hyperactivations. Ulam et al. [29] applied anodal tDCS on the left DLPFC and reported improvements in attention and working memory tests (79% of tests). However, these were also not significant compared to the control group despite the improvement correlating with a decrease in delta waves by electroencephalogram measurements (EEG) [29]. Leśniak et al. [30] also used anodal tDCS on the left DLPFC and reported the intervention group performed better, although not significantly, than the control group in six subcategories of neuropsychological test, the same findings were noted at 4-month follow-up.

#### 3.3.2. Safety

Among five studies included for review, three studies reported no obvious side effects [27,28,29]. Neville et al. [26] reported there was a greater frequency of mild adverse events in the treatment group (70.6% vs. 46.2%, *p* = 0.176) compared to the sham group, though this difference was not statistically significant [26]. Leśniak et al. [30] reported side effects such as tingling (*n* = 6), itching (*n* = 4), drowsiness (*n* = 2), headache (*n* = 1), stinging (*n* = 1), and dizziness (*n* = 1). One patient experienced a panic attack and was consequently excluded from the intervention [30].

## 4. Discussion

We performed a systematic review of the effect of NIBS on cognitive impairment within the TBI population. Based on the review, there is limited evidence of improving cognitive functions such as attention, executive function and working memory using NIBS. However, there were only two studies for rTMS and three for tDCS, which highlights the need for further research to provide additional insights into therapeutic target and stimulation parameters.

DLPFC was selected as the stimulation site in all rTMS and tDCS studies. Four of the five studies [26,27,29,30] selected either excitatory stimulation pattern of the left DLPFC or bilateral DPLFC. Lee et al. [27] selected inhibitory stimulation of the right DLPFC [27] This was based on the inference that activation of the left DPLFC is promoted through interhemispheric inhibition [31]. The reason for choosing DLPFC is the important role that this site exerts in cognitive function. Some studies have shown that DLPFC is associated with attention and working memory function [28,30]. Lee et al. [27] selected the right DLPFC as stimulation site because rTMS for the right DLPFC has positive effect on executive function and depressive symptoms after TBI [27]. Sacco et al. [28] suggested that DLPFC has been found to be involved in dual task processing in healthy subjects. Additionally, there is an increased activity in the DLPFC during working memory tasks [28]. DLPFC is considered a hub of executive function needed to coordinate and integrate different cognitive processes [32]. From fMRI studies, it was observed that cognitive training was strongly associated with reduced involvement of the left DLPFC and increased neural efficiency [33]. As such, selecting of DLPFC as a target in NIBS studies in TBI is logical and consistent with NIBS studies in other illnesses. For example, in a systematic review of the effects of rTMS on Alzheimer’s disease, Liao et al. [34] reported high frequency stimulation (HFS) for right or bilateral DLPFC significantly improved cognition (SMD = 1.06 95%, CI, 0.47–1.66 *p* < 0.05) [34]. Additionally, in a study of NIBS for cognition in Parkinson’s disease, Dinkelbach et al. [35] suggested DLPFC was effective as the stimulation site for both rTMS and tDCS; HFS was effective in rTMS, and anodal stimulation was effective in tDCS [35]. In this review NIBS of the DLPFC was used in all studies.

This review found that two studies reported minor adverse events. In general, The most concerning adverse event with NIBS is the induction of a seizure in the case of rTMS [11] and seizure and skin burn after tDCS [36]. No major adverse events were observed in the current review and no studies reported cognitive deterioration. To establish routine use of NIBS for cognition after TBI, it is necessary to establish a method for identifying the lowest-risk stimulation sites and stimulation parameters. Regarding rTMS after TBI, Li et al. [37] recommends the following to enhance safety: (a) improve accuracy of the stimulation target by using a navigation system and (b) use low frequency rTMS. The navigation system can accurately identify the stimulation site and, in addition, can reduce the propagation of the stimulation to the opposing brain function region. In particular, for patients with large brain lesions, precise setting of the stimulation site by the navigation system may be necessary [38]. LFS can reduce major adverse events such as seizure as compared to HFS. Recently, there have been reports of NIBS using a navigation system after TBI [16,36]. Nielson et al. [39] administered rTMS to a patient with depression who had titanium skull plates inserted following surgery for TBI and reported its efficacy and safety [39]. It is important to note that these case reports utilized LFS, not the previously described excitatory stimulation pattern in DLPFC, which are thought to be effective in improving cognitive function after TBI. It is indicated that more clinical evidence is needed in the future regarding the relationship between safety and stimulation parameters to improve the effectiveness of treatment. To avoid severe side effects when applying excitatory stimulation, it is necessary to consider not only the navigation system described above, but also the use of medication to reduce stimulation threshold, and monitoring of brain imaging via electroencephalogram.

In this systematic review, cognitive rehabilitation and supplementary cognitive training (included computer-assisted training) were conducted in three studies, of which one study in rTMS and one study in tDCS showed improvement [27,28,30]. According to previous reports, NIBS in combination with rehabilitation has demonstrated significant improvements in physical functioning and aphasia after brain injury [31,40]. Restoring impaired neural networks following brain injury is a viable means of promoting functional recovery. In such a situation, a strategy to promote network-related reorganization in the brain must be adopted [41]. NIBS may be a promising complementary treatment when used in conjunction with conventional therapies or cognitive training to enhance rehabilitation in patients with brain injury [15]. From the concept of rehabilitation aimed at improving neuroplasticity, NIBS combined with rehabilitation suggests the possibility of inducing a positive synergistic effect. In addition, this is thought to lead to not only modulation of neural connections, but also functional re-learning. However, the paucity of literature as noted by this review did not provide convincing evidence of the effect of combining rehabilitation with NIBS. More research is needed in the future to review combination therapy with NIBS for cognitive impairment post-TBI.

Based on this systematic review and our previous studies, to build evidence of NIBS for cognitive dysfunction after TBI, it is important not only to evaluate neuropsychological tests, but also to establish evidence for the effects of NIBS itself on neural networks. Regarding the effects of NIBS on neural networks, the mechanism of action differs between rTMS and tDCS, and the mechanism of action of NIBS itself still remains an important debate [12,13]. Previous neuroimaging studies have reported that NIBS affects the cerebral cortex directly under the stimulation site or its functional-related brain regions based on neural networks [42,43,44]. However, consistent is the potential for NIBS to have a positive impact on pathological rhythms in the network post-injury or by disease.

In this systematic review, changes in brain activation were evaluated using neuroimaging and neuropsychological tests [28,29]. Two studies noted that a change in activity in each brain regions was associated with each stimulation site. To make the clinical application of NIBS for cognitive impairment more robust, it is necessary to consider that the site of brain injury varies from patient to patient. The results obtained from NIBS may vary and would be reflected in changes in brain activity using neuroimaging along with neuropsychological tests. Sacco et al. [28] argued that based on their previous study, electroencephalogram (EEG) spectral power measures tracked recovery from TBI in a meaningful way, providing a useful neurobiological marker that could be used to quantify response to rehabilitative interventions, and could potentially become an important predictor of treatment response [28]. As indicated previously, NIBS affects not only the cerebral cortex under the stimulation site but also functional-related brain regions based on neural networks. For example, in a recent study of NIBS for aphasia, it was suggested that the stimulation site and parameter is selected depending on how the damaged language regions and homologous regions related to language acts on the recovery of language function. These selections are based on the duration of onset and the results of changes in brain activity by a language task [45]. In terms of the relationship between NIBS and the effect of neural networks, Padmanbhan et al. [46] reported the relationship between brain function connectivity and post-lesion depression. Lesion locations associated with depression were highly heterogeneous and there were no consistent brain region related to depression. Lesion locations were mapped to a connected brain circuit centered on the left DLPFC; the size of the damaged area alone could predict depression [46]. This same observation may be applied to the relationship between brain lesions and symptoms in cognitive impairment in TBI. Kreuzer et al. [47] also described the relationship and neural connectivity between DLPFC and anterior cingulate cortex. They suggested that rTMS for DLPFC has the secondary effect on anterior cingulate cortex which is functionally related to DLPFC. Therefore, they argued that pre-clinical parameter studies combining rTMS with neuroimaging are necessary [47]. Combining cognitive evaluation with neuroimaging will lead to enhanced evidence of the effectiveness and accuracy of NIBS treatment, and may provide new insights of methods to manage cognitive impairment in TBI populations.

There are some limitations in this review. Firstly, some of the neuropsychological tests included multiple overlapping elements in our symptom-based classification for TBI. As a result, although the correlation between the target symptom improvement and the stimulation site and parameters of NIBS could be assessed, it would be difficult to quantitatively assess these by meta-analysis. Therefore, in NIBS for cognitive impairment for TBI, it is necessary to carefully consider the selection of neuropsychological tests. In addition, neuropsychological tests performed at short intervention intervals can predispose to learning bias. Therefore, in NIBS for cognitive impairment for TBI, it is necessary to carefully consider the selection of neuropsychological tests. Therefore, in order to solve these problems, it is desirable for NIBS for TBI to select the neuropsychological tests that can be evaluated in a short time and that are widely used.

Secondly, in the extracted studies, the target patients were identified based on reported symptoms of cognitive impairment post-TBI as opposed to being classified based on objective measures such as brain imaging or established brain lesion. Understandably, from the standpoint of rehabilitation it is appropriate not to select target patients based on their known lesions. However, for NIBS to be established as a rehabilitation method for cognitive impairment, it is necessary to systematically select patients from their brain function imaging. In addition, it is necessary to perform further evaluations from brain function imaging in order to predict prognosis and identify responders.

Thirdly, the stimulation site was selected only for DLPFC. This is because NIBS for other diseases has a high level of evidence and anatomical brain function within the DLPFC [32,34,35]. This seems reasonable at first glance, but it is possible that this is not all. Therefore, it is necessary to compare it with other stimulation sites. In addition, for rTMS, it is necessary to consider a method that selects a low-frequency stimulation pattern that can affect a wide range of the cerebral cortex [43].

## 5. Conclusions

We performed a systematic review of NIBS on cognitive impairment post-TBI. In all studies, DLPFC was selected as the stimulation site, along with the stimulation pattern promoting activation of the left DLPFC. In some studies, there was a significant improvement compared to the control group, but neither rTMS nor tDCS had sufficient evidence of effectiveness. The paucity of literature on this topic was quite apparent and so more studies are required for the development of treatment parameters and to assess for effectiveness and safety of NIBS for cognitive impairment after TBI. In addition, it is suggested that the neural plasticity change induced by NIBS may contribute to greater improvements when combined with rehabilitation. Finally, evaluation of brain activity at the stimulation site and related areas using neuroimaging on how NIBS acts on the neural network will contribute to the establishment of evidence.

## Figures and Tables

**Figure 1 diagnostics-11-00627-f001:**
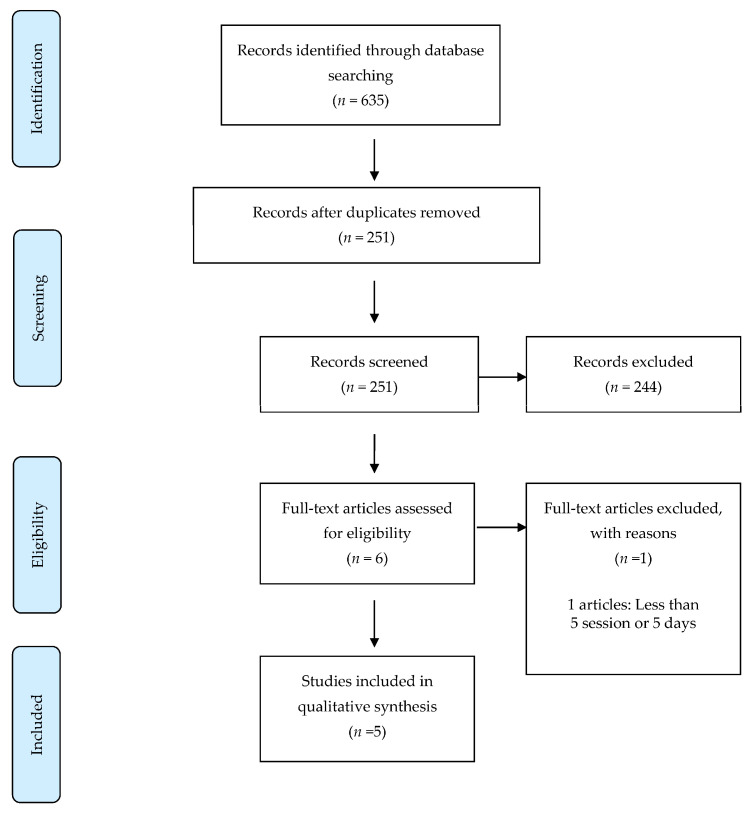
Study flow diagram.

**Table 1 diagnostics-11-00627-t001:** Study characteristics.

Study	Disease	Design-LoE	PEDro	Sample	Sex (M:F)	Age (SD)	Time between TBI Onset and Treatment
TMS							
Neville et al., 2019 [26]	TBI, DAI	RCT-1 I vs. C (Sham)	9	I: 17 C: 13	27:3	I: 29.0 (10.35) C: 32.62 (12.81)	I: 18.30 (13–24) C: 17.62 (13–26) months
Lee et al., 2018 [27]	TBI GCS I: 13.71 (1.11) C: 13.66 (0.81)	RCT-1 I + Re vs. C (Sham) + Re	8	I: 7 C (Sham): 7	9:4	I: 42.42 (11.32) C: 41.33 (11.02)	I: 3.85 (1.67) C: 3.88 (1.94) months
tDCS							
Sacco et al., 2016 [28]	TBI GCS < 8	RCT-2 I + Re vs. C (Sham) + Re	5	I: 16 C (Sham): 16	26:6	I: 37.7 (10.4) C: 35.2 (12.9)	3.16 (17.5) months
Ulam et al., 2015 [29]	TBI	RCT-1 I vs. C (Sham)	8	I: 13 C (Sham): 13	22:4	I: 31.3 (9.8) C: 35.7 (14.7)	I: 57.38 (37.8) C: 41.08 (20.87) days
Leśniak et al., 2014 [30]	TBI	RCT-1 I + Re vs. C (Sham) + Re	7	I: 12 C (Sham): 11	17:6	I: 29.2 (7.3) C: 28.2 (8.6)	18.0 (19.2) months

C = control group, DAI = diffuse axonal injury, GCS = Glasgow Coma Scale, I = Intervention group, PEDro = Physiotherapy Evidence Database, RCT = Randomized controlled trials, Re = Rehabilitation, TBI = Traumatic brain injury.

**Table 2 diagnostics-11-00627-t002:** Individual study treatment characteristics, assessments and outcomes.

Study	Targets	Stimulation Site	Parameter	Session	Rehabilitation	Assessments and Follow-Up	Results
TMS							
Neville et al., 2019 [26]	Attention Memory Executive	Left DLPFC	10 Hz 110% MT 2000 pulses/session	10	None	TMT-A, -B, COWAT, Stroop test, Five-point test, DS SDT, Hopkins verbal learning test, Visuospatial memory test Follow up at 90 days	There was a significant improvement after 90 days in executive function. However, there was no significant difference compared to the control group. No significant differences were observed on other neuropsychological tests
Lee et al., 2018 [27]	Attention Executive	Right DLPFC	1 Hz 100% MT 2000 pulses/session	10	All patients received neurodevelopmental therapy	MADRS, SCWT, TMT	Attention function was significantly improved compared to the control group
Sacco et al., 2016 [28]	Attention Memory WM	Bilateral DLPFC	2 mA/35 cm^2^ × 20 min, Two anodes, one on the right DLPFC and the other on the left DLPFC, earth on the arm	10	All patients received computer-assisted training.	RBANS, BDI, AES Follow-up at 1 month	The intervention group significantly improved in divided attention and attention task of RBANS between before and after treatment. No significant improvement was observed in memory element of RBANS
tDCS							
Ulam et al., 2015 [29]	Cognition Attention Memory WM Executive	Left DLPFC	1 mA/25 cm^2^ × 20 min, Anodal electrode was placed over Left DLPFC and Cathodal electrode placed over the right supraorbital area	10	None	TEA, DS, Symbol span, Color-Word Interference Test, The Awareness of Social Inference Test, Hopkins Verbal Learning Test, The Brief Visuospatial Memory Test	Fifteen out of 19 tests (79%) showed significant pre to post treatment changes. However, no significant difference was observed compared to the control group
Leśniak et al., 2014 [30]	Cognition Attention Memory WM	Left DLPFC	1 mA/10 min/current density = 0.028 mA/cm^2^, Anodal tDCS	15	All patients received rehabilitation program consisted of 15 cognitive training sessions conducted with professional computer software	RAVLT, PRM, RVP, SSP from CANTAB battery, PASAT, EBIQ Follow-up at 4 months	At the post-treatment, the intervention group performed better than the control group in 6 outcome elements. However, none of the differences between groups were statistically significant. At the 4-month follow-up, both groups showed improved performance in most tests. However, the differences between the groups were not sufficiently marked to reach the significance level

DLPFC = Dorsolateral prefrontal cortex, AES = Apathy Evaluation Scale, BDI = Beck’s Depression Inventory, COWAT = Controlled Oral Word Association Test, DS = Digit Span, EBIQ=European Brain Injury Questionnaire, MADRS = Montgomery-Asberg Depression Rating Scale, MT = Motor threshold, PASAT = Paced Auditory Serial Addition Test, PRM = Pattern Recognition Memory from CANTAB battery, RAVLT = Rey’s Auditory Verbal Learning Test, RBANS=Repeatable Battery for the Assessment of the Neuropsychological Status, RVP=Rapid Visual Processing, SCWT = The Stroop Color Word Test, SDT = Symbol Digit Test, SSP = Spatial Span test, TEA = Test of Everyday Attention, TMT = Trail Making Test.

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
