# Peer review of "The Effect of Non-Invasive Brain Stimulation (NIBS) on Executive Functioning, Attention and Memory in Rehabilitation Patients with Traumatic Brain Injury: A Systematic Review"

_diagnostics, 2021, doi:10.3390/diagnostics11040627_

Round 1

Reviewer 1 Report

Authors examine an interesting topic regarding the use of non-invasive brain stimulation for improvement of cognitive deficits post-TBI. However, there are some major flaws that precludes publication at this stage:

1) The PROSPERO registration number refers to another protocol from the same group (with the same topic but in the context of stroke rehabilitation)- this should be updated with the correct ID before this study can be formally assessed as a systematic review

2) The number of studies and sample sizes appear too small to perform any meaningful meta-analysis; if this paper were to be published, it would make more sense to re-structure it as a scoping review or SWiM

Minor issues

1) Results section mentions 6 studies eligible for inclusion whilst the abstract and figures only mention 5 studies

2) Tables could be improved in terms of readability (e.g. is a Disease column required in Table 2 in a systematic review about NIBS in TBI ?)

3) Was PEDro used in anyway to discriminate studies eligible for meta-analysis ? In either case, see major flaw (2)

Author Response

Reviewer 1

Thank you for your insightful feedback, we have now adjusted our manuscript to address the concerns raised, we hope this would improve our manuscript to make it acceptable for publications. Please see answers to the points you raised below.

1. The PROSPERO registration number refers to another protocol from the same group (with the same topic but in the context of stroke rehabilitation). This should be updated with the correct ID before this study can be formally assessed as a systematic review.

The initial registration with PROSPERO was for the protocol to review NIBS for stroke and TBI. While we did perform the review according to this protocol, we have opted to publish the results for each neurorehabilitation population (stroke vs. TBI) separately, therefore, the number remains the same. We have clarified this in the manuscript.

2. The number of studies and sample sizes appear too small to perform any meaningful meta-analysis; if this paper were to be published, it would make more sense to re-structure it as a scoping review or SWiM.

We agree that there is little to be gained by performing a meta-analysis given the few studies retrieved from the literature search, so we have opted to remove this aspect of the study; however, we believe a synthesis of the evidence without a meta-analysis (systematic review) is still necessary and the text has been revised.

3. Results section mentions 6 studies eligible for inclusion whilst the abstract and figures only mention 5 studies.

Thank you for highlighting this error. We have corrected the text and citation.

4. Tables could be improved in terms of readability (e.g. is a Disease column required in Table 2 in a systematic review about NIBS in TBI?)

Thank you for your advice; we have modified the Tables for improved readability.

5. Was PEDro used in anyway to discriminate studies eligible or meta-analysis? In either case, see major flaw (2)

The meta-analysis has been removed. Methodological quality of each study was assessed using PEDro and reported in Table 1.

Reviewer 2 Report

This systematic review of the literature (English publication) concerning Non Invasive Brain stimulation -(both TMS and T-DCS) examined the studies to evaluate the effects of these modalities on attention, memory and executive function in those with traumatic brain injury.  The selection criteria used to choose those studies to be included in the analysis was clearly stated and seems clinically relevant and appropriate.

It was also clear the the brain stimulation was commonly used as an adjunct to the more accepted tools for cognitive rehabilitation.

In examining the small number of studies (5 in total - 2 with TMS and 3 with tDCS lead to a total N of 135 subjects )

Positive impact on attention and memory was measured in 2 of the three studies for tDCS, but it is clear the changes seen were not statistically significant with respect to their appropriate control groups.

Using this analysis of the literature it seems clear there is no lasting or significant effect of TMS or tDCS on cognitive function in TBI rehabilitation that yet can be seem.  I would state that at present the conclusion is that NIBS cannot be seen as effective in the rehabilitation of Cognitive function in TBI, rather than the conclusion that the field is "in its infancy". 

It is clear that the rehabilitation literature in this area of field is still lacking depth and breadth.  As ALWAYS well designed and conducted studies with clinical power  are needed to answer these clinically relevant questions 

Author Response

Reviewer 2

Thank you for your insightful feedback, we have now adjusted our manuscript to address the concerns raised, we hope this would improve our manuscript to make it acceptable for publications. Please see answers to the point you raised below.

[…] I would state that at present the conclusion is that NIBS cannot be seen as effective in the rehabilitation of Cognitive function in TBI, rather than the conclusion that the field is "in its infancy". It is clear that the rehabilitation literature in this area of field is still lacking depth and breadth. As ALWAYS well designed and conducted studies with clinical power are needed to answer these clinically relevant questions.

Thank you for your suggestion; we have revised the text to be clearer in our abstract, limitations, and conclusions.